# Drill site selection for cosmogenic nuclide exposure dating of the bed of the Greenland Ice Sheet

Jason P. Briner[1][*], Caleb K. Walcott[1], Joerg M. Schaefer[2], Nicolás E. Young[2], Joseph A. MacGregor[3], Kristin Poinar[1], Benjamin A. Keisling[2], Sridhar Anandakrishnan[4], Mary R. Albert[5], Tanner Kuhl[6] and Grant Boeckmann[6]

*1 Department of Geology, University at Buffalo, Buffalo, NY, 14260 USA*

*2 Lamont-Doherty Earth Observatory, Columbia University, Palisades, NY, USA*

*3 Cryospheric Sciences Laboratory, NASA Goddard Space Flight Center, Greenbelt, Maryland, USA*

*4 Department of Geosciences, Penn State University, University Park, PA 16802, USA*

*5 U.S. Ice Drilling Program, Thayer School of Engineering, Dartmouth College, Hanover, NH, USA Ice Drilling Program*

*6 U.S. Ice Drilling Program, University of Wisconsin-Madison, Madison, WI, USA*

*\*Corresponding author, jbriner@buffalo.edu*

## 1. Abstract

Direct observations of the size of the Greenland Ice Sheet during Quaternary interglaciations are sparse yet valuable for testing numerical models of ice sheet history and sea level contribution. Recent measurements of cosmogenic nuclides in bedrock from beneath the Greenland Ice Sheet collected during past deep drilling campaigns reveal that the ice sheet was significantly smaller, and perhaps largely absent, sometime during the past 1.1 million years. These discoveries from decades-old basal samples motivate new, targeted sampling for cosmogenic nuclide analysis beneath the ice sheet. Current drills available for retrieving bed material from the US Ice Drilling Program require <700 m ice thickness and a frozen bed, while quartz-bearing bedrock lithologies are required for measuring a large suite of cosmogenic nuclides. We find that these and other requirements yield only ~3.4% of the Greenland Ice Sheet bed as a suitable drilling target using presently available technology. Additional factors related to scientific questions of interest are which areas of the present ice sheet are the most sensitive to warming, where a retreating ice sheet would expose bare ground rather than leave a remnant ice cap, and which areas are most likely to remain frozen bedded throughout glacial cycles and thus best preserve cosmogenic nuclides? Here we identify locations beneath the Greenland Ice Sheet that are best suited for potential future drilling and analysis. These include sites bordering Inglefield Land in northwestern Greenland, near Victoria Fjord and Mylius-Erichsen Land in northern Greenland, and inland from the alpine topography along the ice margin in eastern and northeastern Greenland. Results from cosmogenic nuclide analysis in new sub-ice bedrock cores from these areas would help to constrain dimensions of the Greenland Ice Sheet in the past.

## 2. Introduction

Recent observations reveal significant ice loss in Greenland and Antarctica, with the Greenland Ice Sheet (GrIS) presently contributing more to sea level rise than the Antarctic Ice Sheet (AIS) (Shepherd et al., 2018; Shepherd et al., 2020). The higher potential for portions of the AIS to collapse due to marine ice-sheet instability, however, leaves estimates of future sea

level rise highly uncertain (Scambos et al., 2017; DeConto et al., 2021; Edwards et al., 2021).
Non-linearities in ice sheet response to climate change also apply to the GrIS, which has been
simulated to disappear in as little as one millennium (Aschwanden et al., 2019). Estimated rates
of GrIS loss this century under the current trajectory of greenhouse-gas emissions (Goelzer et
al., 2020; Edwards et al., 2021) have been shown to exceed those under natural variability over
the past 12,000 years (Briner et al., 2020).

Although present rates of ice sheet loss are exceptional and concerning, there are few

direct constraints on GrIS and AIS response to similar warmth during past interglaciations of the
Quaternary (e.g., deVernal and Hillaire-Marcel, 2008; Schaefer et al., 2016). Thus, knowledge of
ice sheet response under past climates that are comparable to the climate of our near future
remains limited. Proxy data from geological archives, such as sedimentological characteristics in
adjacent seas, have been used to evaluate ice sheet history, although these provide indirect
evidence only for past changes in ice-sheet size. A growing body of evidence from offshore
Greenland documents overall ice sheet growth and its subsequent oscillatory configurations
throughout the Pliocene and Quaternary (e.g., Bierman et al., 2016, Knutz et al., 2019).
Paleoceanographic studies have made valuable inferences of climate conditions (e.g., de Vernal
and Hillaire-Marcel, 2008; Cluett and Thomas, 2021) and ice sheet configuration (e.g., Reyes et
al., 2014; Hatfield et al., 2016) during brief interglacials. Generating direct knowledge of past
GrIS response to interglacial warmth has proven difficult with these approaches. Farfield sea
level reconstructions help to constrain GrIS response during past interglaciations (e.g., Dyer et
al., 2021), yet still benefit from direct observations from individual ice sheets. Ice sheet
modeling has simulated a variety of ice sheet volumes and configurations during past
interglaciations (e.g., Goelzer et al., 2016; Robinson et al., 2017; Plach et al., 2018; Sommers et
al., 2021), indicating more geologic measurements of ice-sheet extent are needed to evaluate
these results.

The age of ice in basal ice core sections has been used to constrain the GrIS

configuration during marine isotope stage (MIS) 5e (129-116 ka) and thus validate numerical
simulations of ice size and configuration during the last interglacial (Otto-Bleisner et al., 2006;
Plach et al., 2018; Domingo et al., 2020). However, there is some uncertainty about the role
that ice advection plays in bringing aged ice over a previously ice-free location. For example,
Yau et al. (2016a) found that the best fitting models for matching their elevation and
temperature reconstructions for NEEM and GISP2 did not have ice at NEEM during MIS 5e,
implying that the MIS 5e ice recovered at NEEM today not only flowed laterally but re-advanced
over a deglaciated landscape. This phenomenon can be observed directly at the modern ice-
sheet margin today, where Pleistocene-age ice outcrops at the margin in western Greenland
(e.g., Reeh et al. 2002; MacGregor et al. 2020) where there was no ice as recently as the middle
Holocene (Briner et al., 2010). Thus, it is critical to obtain independent information about sub-
ice bedrock exposure age because apparently the age/stratigraphy of the overlying ice does not
necessarily provide a continuous constraint on ice-cover history.

Fortunately, a new frontier of science is emerging, aimed at generating direct

constraints on former ice sheet size using information collected from the ice sheet bed.
Schaefer et al. (2016) measured cosmogenic $^{10}$Be and $^{26}$Al in bedrock obtained below the GISP2
ice core (Figure 1), equipped with updated procedures and vastly improved analytic sensitivity
relative to an earlier attempt (Nishiizumi et al., 1996). Their measurements require the GrIS to
have been absent at the GISP2 locality for 280 kyr of the past 1.4 Myr. Although alternative
histories are possible, the results lead to an important conclusion: an almost entirely absent ice
sheet in Greenland within the last 1.1 Myr. Furthermore, these types of data directly constrain
past ice-sheet configurations, unlike marine sediment records from adjacent seas that provide
indirect evidence. More recently, Christ et al. (2021) measured cosmogenic $^{10}$Be and $^{26}$Al in re-
discovered sub-ice sediments in the Camp Century ice core collected in the 1960s (Figure 1).
They interpret their results to indicate that the landscape below Camp Century became ice free
at least once in the last 1.0 Myr. While one might expect the GrIS flank site of Camp Century to
become ice free during some interglacial periods (model simulations commonly show this;
Plach et al., 2018; Sommers et al., 2021), the findings from beneath the summit of the GrIS
were more unexpected because model simulations rarely show ice-free conditions there (Briner
et al., 2017). Additionally, new approaches have been developed to solve for long-term ice
sheet occupation and subglacial erosion histories from vertical profiles of cosmogenic nuclides
measured in multiple meters of rock core (Balter-Kennedy et al., 2021). Performing such
analyses on new multi-meter-long bedrock cores from beneath the GrIS will be key for
deciphering GrIS history.

Cosmogenic-nuclide measurements from sub-ice bed material in Greenland already

have been shown to place direct constraints on past ice sheet history, despite the study of only
two cosmogenic isotopes ($^{10}$Be and $^{26}$Al) in these samples thus far. Additionally, the recent
results from the sub-GrIS environment, although derived using legacy material from sites not
targeted for cosmogenic-nuclide measurements, have demonstrated the power of this
approach. While drilling technology that allows quick access (i.e., in a single field season) to the
bed below ice sheet summits is being developed for application in Antarctica (Goodge and
Severinghaus, 2016; Goodge et al., 2021), there is no such drill – or plans for one – to operate in

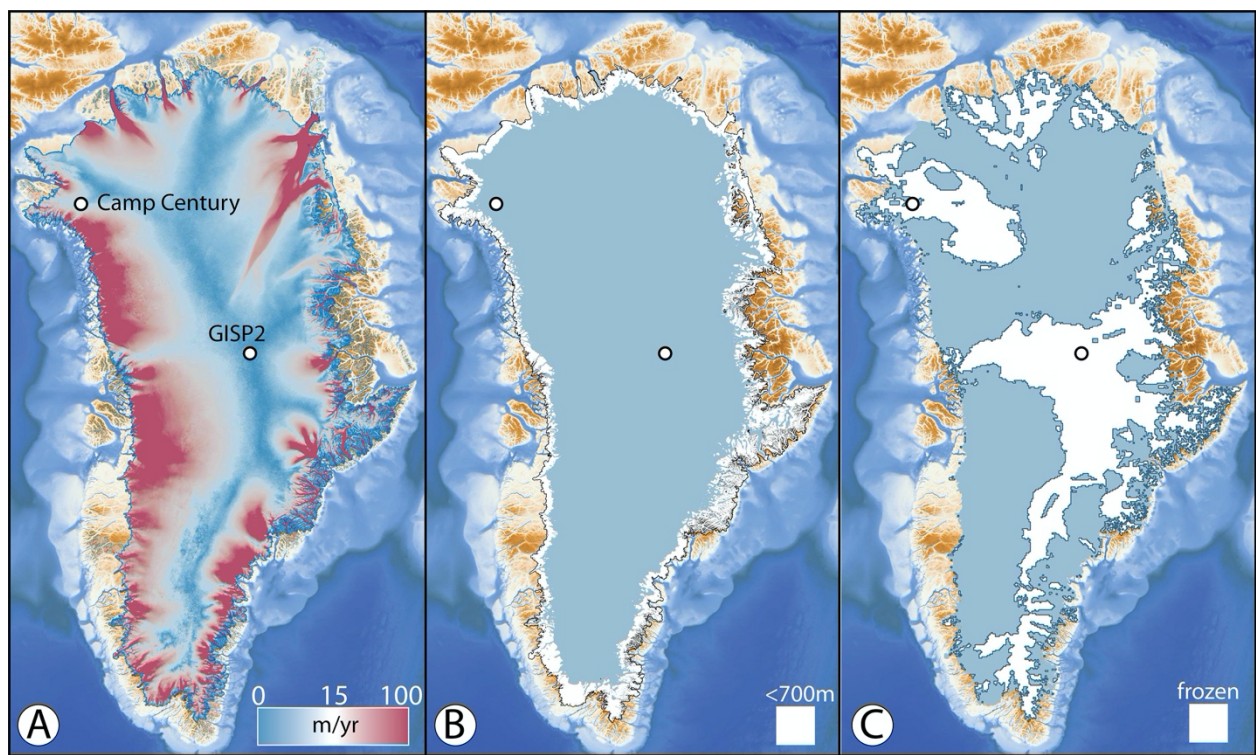

**Figure 1.** A. Horizontal surface velocities of the Greenland Sheet; the slowest flowing-areas define the summit ridge and ice divides; velocity from Greenland Ice Sheet velocity map from Sentinel-1, winter campaign 2019/2020 [version 1.3]; QGreenland v2.0. B. The pattern of <700 m ice thickness (white) shown around perimeter of the ice sheet, which covers 15.2% of the ice-sheet footprint. C. Where the basal thermal state is likely frozen bedded (white), which covers 37.4% of the ice-sheet footprint (from MacGregor et al., 2022). Basemap topography and bathymetry from Morlighem et al. (2017).

Greenland. However, there are drills designed to quickly access the bed in locations where ice
thickness is <700 m (Spector et al., 2017, 2018). The goal of this study is to survey Greenland to
identify sites that are potentially suitable for sub-ice cosmogenic-nuclide measurements using
two suitable drills in the US Ice Drilling Program's inventory: the Agile Sub-Ice Geological (ASIG;
Kuhl et al., 2021) drill and the Winke Drill (Boeckmann et al., 2021). Both of these drills can
operate in Greenland. Considering drill specifications, scientific and safety criteria, we identify
multiple suitable sites near the GrIS margin across northern and eastern Greenland. These sites
represent candidate targets for the GreenDrill project supported by the U.S. National Science
Foundation.

**3 Considerations for drilling**
The drills currently available from the US Ice Drilling Program that are designed to drill
rock cores beneath tens to hundreds of meters of glacial ice require the bed beneath the ice to
be frozen to its bed. Additional specifications for scientific projects focused on sub-ice samples
obtained via drilling, such as bedrock lithology and site accessibility, further limit suitable areas.
The bedrock lithology of Greenland is varied and is only exposed around the island's perimeter
and directly observable in only one hand-sample from the base of the GISP2 ice core site. With
only six locations across the GrIS interior where boreholes have reached the bed, there are also
limited direct observations of the ice sheet's basal thermal state. Below, we compile this and
other necessary information for identifying potential sites for retrieval of rock cores beneath
the GrIS.

*3.1 Drills*
We first briefly outline the technical requirements of the two presently available US Ice
Drilling Program drills designed to drill through ice and into the underlying bedrock: ASIG and
Winkie (Albert et al., 2020). The ASIG Drill is currently designed to drill access holes through ice
<700 m thick and collect bedrock cores several meters long. It requires frozen basal conditions
to ensure that drilling fluid is maintained in the entire borehole across the ice–bed interface.
The ASIG drill was successfully used in West Antarctica near the Pirrit Hills in 2016-2017, where
it drilled through approximately 150 m of ice and collected 8 m of 39-mm-diameter rock core of
excellent quality (Kuhl et al., 2021). Nearly 5 m of ice core was also collected near the ice-
bedrock transition, however, the core quality was poor. The Winkie Drill is capable of drilling
120 m of ice and rock (e.g., it can retrieve a 10 m rock core from beneath 110 m of ice); it also
has the requirement of a frozen bed. Given these restrictions, the Winkie Drill is mostly
restricted to frozen-bedded environments very near the GrIS margin, and the ASIG Drill is
suitable to drill in similar environments slightly farther inland.

### 3.2 Ice thickness
The large-scale thickness of the GrIS is relatively well known, stemming from several
decades of radar data collection by NASA and European institutions (e.g., Li et al, 2012).
Morlighem et al. (2017) combined airborne radar-sounding-derived ice thickness data with
comprehensive, high-resolution ice motion measurements derived from satellite
interferometric synthetic-aperture radar. This combination of datasets allowed Morlighem et
al. (2017) to employ a mass conservation algorithm (Morlighem et al., 2011; McNabb et al
2012) to calculate ice thickness around the periphery of the GrIS. They produced a map of bed
topography by subtracting ice thickness from a digital elevation model of the ice surface. Mass
conservation works best in areas of fast flow, where uncertainty in flow direction is small and
the glaciers mostly flow due to basal motion (Morlighem et al., 2011). In the interior, where
deformation is likely a more dominant component of ice flow and uncertainty in flow direction
is greater, they employed ordinary kriging to interpolate ice thickness measurements. We use
BedMachine v3 (Morlighem et al., 2017) and ArcGIS to deduce that 15.2% of the GrIS is <700 m
in thickness (Figure 1B).

### 3.3 The basal thermal state of the GrIS
Due to the limited number of boreholes that have reached the GrIS bed, its basal
thermal state must presently be estimated from a synthesis of multiple methods. MacGregor et
al. (2016, 2022) combined thermomechanical ice-flow models and inferences from airborne
and satellite remote sensing to constrain where the bed is likely thawed, where it is likely
frozen and where it remains too uncertain to specify, at a spatial resolution of 5 km. The latest
version of this synthesis of the GrIS likely basal thermal state (MacGregor et al., 2022) is shown
in Figure 1C. The map suggests frozen-bedded conditions across 37.4% of the ice sheet, mostly

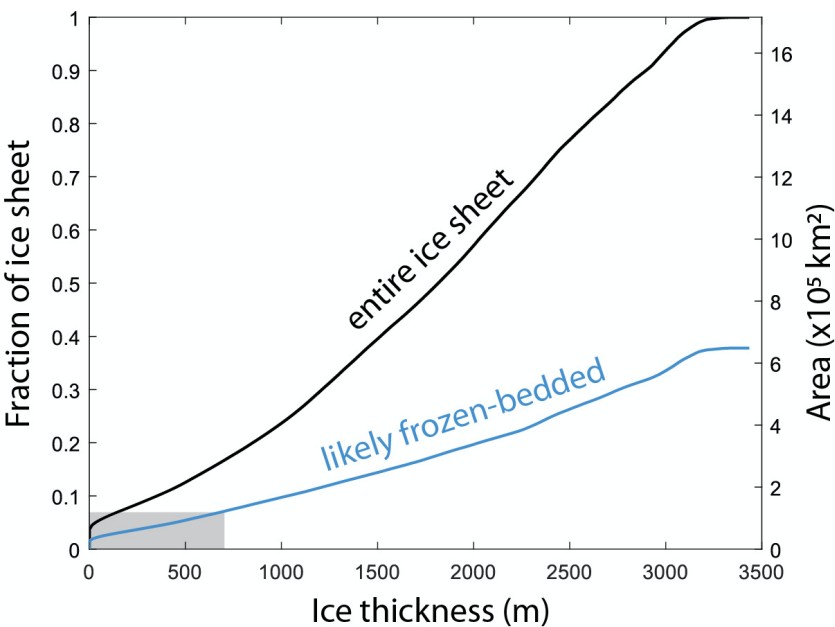

**Figure 2.** Greenland Ice Sheet thickness versus area. Plot shows that only about one-third of the ice sheet by area is likely frozen-bedded, and thus available for subglacial access. The current limit of drills of operating under 700 m ice thickness further reduces available portion of the ice-sheet bed for access (gray area). Note that increasing a drill's depth capability increases the area of the bed available for drilling; the ability to drill into wet-based sections of the ice sheet would significantly increase the area available for drilling.

beneath ice divides and parts of North Greenland (Figure 2). The ice margin and near-ice-
margin areas throughout most of Greenland are largely believed to be thawed, except for a few
locations across North and East Greenland where frozen-bedded conditions are ubiquitous –
even near the ice margin. However, there are many areas where the basal thermal state is
mapped as uncertain (i.e., areas that are inconclusive in terms of their likelihood to be either
warm-or frozen-bedded), and many of these areas also extend to the ice margin in portions of
North and East Greenland. Jordan et al. (2018) used radar returns to identify locations of
probable water at the bed. Although the method could not be applied throughout Greenland
due to limitations in radar extent and quality, their fine-resolution dataset was included by
MacGregor et al. (2022). Bedrock weathering textures and landforms observed in landscapes
occupied by Pleistocene ice sheets reveal sharp transitions between warm- and frozen-bedded
conditions in the past, particularly in areas of high topographic relief (e.g., Sugden, 1978; Briner
et al., 2006). Thus, there could be localized patches of frozen-bedded conditions across many
areas around the GrIS perimeter that are too small in scale to be suitably represented using the
methods of Jordan et al. (2018) and MacGregor et al. (2022). Combining likely frozen basal
conditions with ice thicknesses <700 m results in 6.8% of the bed available for drilling (Figures 2
and 3A).

Finally, prior to drilling, the selected sites should be assessed with geophysical methods

to further estimate the thermal state of the bed. Existing radar profiles combined with new
radar and seismic measurements can reduce the uncertainty about the condition of the bed.
Seismic methods can more confidently measure whether a significant water volume is present
at the bed, either pooled or within sediment pores (e.g., Kulessa et al., 2017). The reflectivity of
water or water-laden sediments is significantly different than for frozen sediments. Note that a
thin layer of water over crystalline bedrock would be difficult to distinguish from frozen ice over
bedrock.

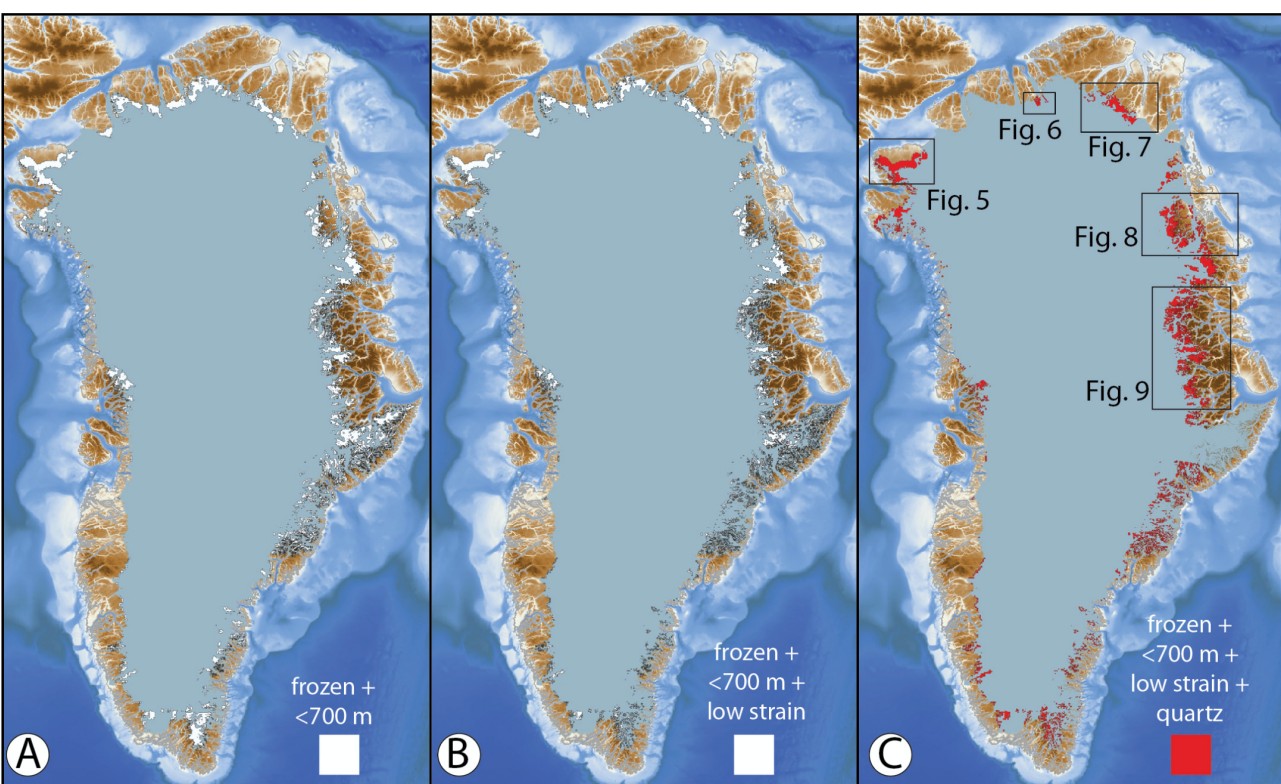

**Figure 3.** A. Portion of the Greenland Ice Sheet bed (6.8%) that are both likely frozen and beneath <700 m of ice. B. Same as in A but which has a low likelihood of surface crevasses (4.8%). C. Same as in B but with likely quartz-bearing lithologies (3.4%). Basemap topography and bathymetry from Morlighem et al. (2017).

*3.4 Surface features, safety and site accessibility*
Because available drills require <700 m ice thickness, the viable areas of interest are
mostly restricted to near the ice margin (Figure 1B). These areas generally have high surface
velocity (>50 m/yr) and spatial variability in surface velocity as ice flow becomes increasingly
influenced by underlying topography (Figure 1A). Consequently, these areas have high strain
rates and can be heavily crevassed, making them some of the most dangerous locations on the
GrIS to work. However, not all ice-marginal areas exhibit high velocity and high strain rates, so
some areas are relatively crevasse free. Surface strain rates derived from GrIS surface velocity
(Figure 1A) can guide site selection for low likelihood of crevassing. In this way, one can address
the criterion of being most likely to be safe for air support and/or access via traverse vehicles.
We use a strain-rate field from Poinar and Andrews (2021) and a threshold value of 0.005/year,
above which crevasses are likely to form (Joughin et al., 2013). This analysis further reduces the
area of the GrIS suitable for drilling from 6.8% to 4.8% (Figure 3B).

***3.5 Cosmogenic nuclides and subglacial geology***
An entire family of cosmogenic nuclides are routinely measured in Earth materials. Most
research to date in Earth science, however, has used cosmogenic nuclides produced in quartz:
$^{26}$Al, $^{14}$C and $^{10}$Be (Granger et al., 2013; Briner et al., 2014; Balco, 2020). While there are
cosmogenic nuclides that can be used in mafic lithologies (e.g., $^{36}$Cl, $^{3}$He) and carbonates (e.g.,
$^{36}$Cl), the advantage of quartz is that the trio of $^{26}$Al, $^{14}$C and $^{10}$Be can all be measured together
(e.g., Young et al., 2021). These three nuclides have widely spaced half lives, providing a
powerful exposure-burial chronometer well suited for providing direct constraints on ice sheet
history. Additionally, $^{36}$Cl can also be measured in feldspars, and thus targeting felsic-crystalline
lithologies potentially offers a fourth cosmogenic nuclide with a unique half-life for analysis.
Because 81% of Greenland's land area lies beneath the ice, bedrock geology has only
been mapped across 19% of Greenland. There is a large degree of uncertainty about the
lithology below the ice sheet. Dawes (2009) inferred the sub-GrIS geology based on information
from six methods: Drill sites, nunataks, coast-to-coast correlation, glacial erratics, detrital
provenance studies and geophysics. For the purpose of identifying sites for cosmogenic-nuclide
analysis, we provide a highly abbreviated overview of this geology with particular attention paid
to quartz-bearing lithologies in areas likely to coincide with frozen-bedded conditions. We use
the geologic map of Greenland, available online at https://www.greenmin.gl/ (Pedersen et al.,
2013; Henriksen et al., 2009). Generally, Greenland mostly consists of Precambrian shield rocks
(both Archean and Proterozoic; largely quartz-bearing) in its southern, western and central
areas. North Greenland consists of Paleozoic basins containing mostly non-quartz-bearing
lithologies. East and Northeast Greenland comprise the Caladonian fold belt and a complex
pattern of Proterozoic rocks of mixed lithology, although these are thought to be mainly limited
to the island's periphery. Portions of the central east and central west coasts of Greenland
contain Paleogene volcanic lithologies that may connect beneath the central GrIS. North
Greenland generally encompasses the highest proportion of the margin and near-margin areas
thought to be frozen-bedded; however, carbonate and other non-quartz-bearing lithologies
dominate these areas. We use the geologic map of Greenland to categorize bedrock lithology
into quartz-bearing and non-quartz-bearing units (Figure 4). We remove the ice-marginal areas
adjacent to carbonate and volcanic lithologies from consideration, which reduces the target
area from 4.8% to 3.4% of the GrIS (Figure 3C).
Cosmogenic nuclide analyses made in a depth profile below the ice-bed interface yields
important information. Measurements in a rock core spanning a meter or more, for example,
can allow one to easily identify whether or not the current ice-bed interface has been eroded
and/or covered by snow, ice or sediment for long durations (e.g. Schaefer et al., 2016). On the
other hand, one cannot determine with surface-only samples whether a surface has been
impacted by minor erosion and/or burial by snow, ice or sediment. Thus, analysis of bedrock
cores is most important for elucidating ice sheet histories from cosmogenic-nuclide inventories.
Furthermore, cores spanning several meters and including depths dominated by muon
production have the added advantage of constraining orbital-scale term exhumation histories
(e.g., Balter-Kennedy et al., 2021).
Sampling from a bedrock substrate has advantages over samples from sediment
deposits, although cosmogenic nuclide measurements from both are informative. Sediments
beneath ice sheets are more easily eroded, deformed, entrained, transported and re-deposited
than bedrock. Thus, cosmogenic nuclide concentrations from the sediment grains themselves,
which have a transport and deposition history, are more complicated to interpret than those in
bedrock. Furthermore, cold-based ice that flows atop sediment sections can more easily erode
a sediment surface (via entrainment processes) than in bedrock substrates. Thus, not only is a
cosmogenic signal in sediments derived from each individual grain's exposure and burial
elsewhere (that are later amalgamated into a single deposit), but the ice-bed itself may not
represent a prior land "surface." Thus, sediment samples could be from an arbitrary depth
below a paleo-surface. The depositional environment of sediment is also important. If ice
overlies a fluvial sediment sequence, then the cosmogenic nuclide inventory is highly likely to
have a complicated genesis, and thus a more complicated interpretation. On the other hand, if
the sediment is saprolite or regolith, and largely formed in-situ, then its cosmogenic nuclide
inventory likely would be more straight forward to interpret. In any case, for targeted sub-GrIS
cosmogenic nuclide campaigns, the highest priority sites are those where non-erosive ice rests
directly on quartz-bearing bedrock.

How to maximize the chance of drilling into bedrock? Site selection is aided by airborne

radar sounding data obtained by NASA Operation IceBridge. Existing surveys of the ice sheet
bed are inadequate for identifying every low topographic swale that could potentially be
sediment filled, particularly between radar flight lines. However, by avoiding valleys and low
areas and instead opting for mountain summits or plateaus, we can increase the likelihood of
drilling into bedrock with thin or no sediment cover. Although not always the case, in most
areas of Greenland that are ice free today, bare bedrock surfaces generally exist in higher
proportion on hilltops and uplands, as opposed to low-lying areas and valley bottoms. Thus,
choosing sites along radar flight lines ensures the most reliable knowledge of bed topography
and ice thickness at a candidate drill site.

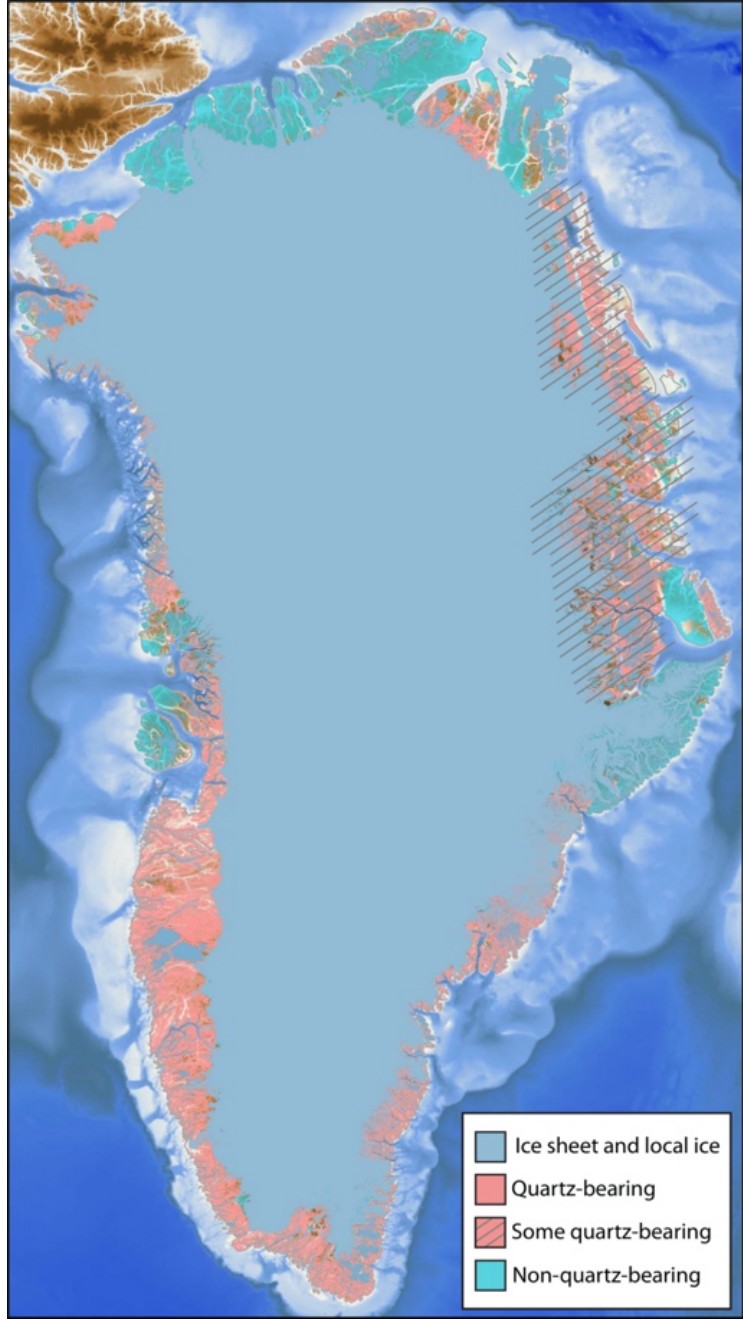

**Figure 4.** Simplified bedrock geology map of Greenland, showing lithology sub-divided into probably quartz-bearing rocks, some quartz bearing lithologies and probably non-quartz-bearing rocks. From https://www.greenmin.gl/. Basemap topography and bathymetry from Morlighem et al. (2017).

***3.6 Strategizing drill site selection related to scientific motivation***

Having applied above the drilling and geologic requirements for site selection, we next

consider the scientific progress that could be realized from the analysis of bed materials at a
particular site. With the goal of constraining Pleistocene GrIS history in mind, we consider four
primary criteria.

First, the best sites should be robust monitors of past ice-sheet margin change. There

could be regions, such as high-elevation terrain (e.g., in mountainous East Greenland) that meet
the technical criteria but retain local ice cover during times of reduced ice-sheet configurations,
complicating the link between the study site and broader GrIS change. There could also be sites
that are part of the GrIS but are better conceived of as separate ice domes connected to the ice
sheet via a saddle; these 'local' domes may persist longer than the adjacent ice sheet during
interglacial periods as disconnected ice caps (e.g., Prudhoe Dome, Figure 5). Ice-sheet modeling
should be used to help guide site selection (Keisling et al., 2022).

Second, to best capitalize on new measurements of cosmogenic nuclide signatures of

past ice sheet changes (Spector et al., 2018; Keisling et al., 2022), sites should be sought that
have persistent frozen-bedded conditions throughout glacial–interglacial cycles. These sites
should favor preservation of cosmogenic nuclides at the ice–bed surface and reduce the
likelihood of significant periods of time with subglacial erosion that removes the cosmogenic
nuclide inventory. At sites where sub-glacial conditions favor erosion, cosmogenic nuclides
would be largely absent, hence removing one of the main reasons for obtaining sub-ice bedrock
samples. Identifying these sites could be based on a combination of selecting presently frozen
bedded areas, favoring high-elevation locations or ice divide areas likely to be frozen bedded
during past larger ice-sheet configurations, and evaluating paleo ice-sheet models to find ideal
drilling locations.

Third, there may be some sites that are more sensitive monitors of reduced ice extent

than others. For example, while some sites at 600 m ice thickness today may become ice free at
under 5% reduction in ice-sheet mass, others may not become ice-free until a substantially
greater reduction in mass. Using numerical ice-sheet models could greatly assist site selection
and help to further explore sites that meet the technical requirements for their potential to
constrain past ice sheet configurations (Keisling et al., 2022).

Fourth, some ice-sheet margin areas that include large, fast-flowing outlet glaciers with

beds below sea level (e.g., near Jakobshavn Isbræ, Petermann Glacier, Northeast Greenland Ice
Stream), could potentially 'collapse' at rates faster than other ice sheet margin areas. Thus,
sites neighboring these regions, such as the Northeast Greenland Ice Stream, could not only
serve as a binary signal of ice sheet presence/absence, but could help to elucidate the response
of major outlet glaciers influenced by ice-ocean interactions to past climate forcing. Does the
Northeast Greenland Ice Stream collapse during past warm times and exhibit proportionally
more ice sheet recession than other ice-sheet sectors?  Sites adjacent to the Northeast
Greenland Ice Stream could help resolve this question.

Finally, additional considerations relating to field season planning could lead to meeting

scientific goals most efficiently. Multiple drill cores along transects (even including sites beyond
the present – ephemeral – position of the ice-sheet margin) could boost confidence in
constraining past ice-sheet dimensions through time. For example, a site that is presently
covered by 100 m of ice may have been ice-free during the Holocene, whereas a 400-m-thick
site farther inland was not; thus, one could better constrain the position of the ice margin
during the middle Holocene. Additionally, using the ASIG Drill, there may be enough time in a
field season to acquire one or two drill cores from thicker ice sites (e.g., 500-700 m), versus
obtaining many drill cores in a single season from ~100m-thick sites using the Winke Drill. The
optimal sampling strategy depends on several factors that relate to a particular scientific
objective.

**4. Areas suitable for drilling using ASIG**

We synthesized the information discussed above to derive a map of candidate areas

across the GrIS for drilling (Figure 3C). While only 3.4% of the GrIS bed is well suited for
subglacial access for the purpose of cosmogenic-nuclide analysis, there are several promising
candidate sites: (1) Northwest Greenland, specifically the metamorphic lithologies of the
Ellesmere-Inglefield Province in Prudhoe Land-Inglefield Land; (2) Two regions in North
Greenland: a small area around the head of Victoria Fjord that likely exposes metamorphic
lithologies of the Victoria Province and an area adjacent to Mylius-Erichsen Land in eastern
North Greenland that contains siliciclastic sedimentary units; (3) Dronning Louise Land in
Northeast Greenland, where both crystalline and siliciclastic lithologies are present; and 4)
central East Greenland, where the GrIS flows through alpine terrain of mixed lithology en route
to the headwaters of the Scoresby Sund, Kong Oscar Fjord and Kejser Franz Joseph Fjord
systems. There are additional small areas scattered around the periphery of the GrIS; however,
most areas are in alpine-style, icefield-type settings, or lie in small areas between outlet
glaciers. In these additional small areas, however, existing uncertainties in available datasets
(e.g., ice sheet thickness, basal temperature, etc.) means that drilling there is potentially riskier
than in larger patches of the bed that meet drilling requirements.

**4.1 Northwest Greenland: Prudhoe Land and Inglefield Land**

In Northwest Greenland, the ice sheet in Prudhoe Land and Inglefield Land has broad

areas that meet the technical, safety and lithology criteria (Figures 3C and 5). Here, there are
basement rocks consisting of Proterozoic metamorphic lithologies. Proportionally much of the
region contains quartz-bearing metamorphic rocks (e.g., paragneiss), albeit with varying quartz
content, and in some cases with bands of marble and other potentially non-quartz-bearing
units (e.g., syenite, amphibolite; Henriksen et al., 2009). Further, the ice sheet has been
surveyed extensively by NASA's Operation IceBridge and abundant radar data exist. The ice
sheet margin adjacent to Inglefield Land, spanning between Hiawatha Crater and Prudhoe
Dome, is roughly parabolic in profile and rather uniform in velocity, with surface speeds mostly
ranging from 3–10 m yr$^{-1}$. The topography of the ice sheet bed is low-relief, potentially making
it difficult to identify small hills and swales where the substrate is less or more likely to host
sediment. The landscape fronting the ice is largely bedrock, or bedrock overlain by surface
blocks either frost-riven or slightly modified by former glaciation. In a few areas, alluvium or
glacial deposits exist at the surface. Prudhoe Dome itself (Figure 5) has a thickness of ~500 m at
a summit ridge that rests along a topographic high above a bed elevation of ~800 m asl. The
velocities in the summit region of Prudhoe Dome range up to ~20 m yr$^{-1}$. The Prudhoe Dome
summit is a promising place to drill, with a high probability of encountering bedrock at the ice
sheet bed. However, upon deglaciation, the site may maintain local ice isolated from the inland
ice, potentially fueled by snowfall due to its proximity to Baffin Bay. Ice sheet modeling could
be used to detect inland (ice sheet) versus peripheral (local) ice survival in locations like this.

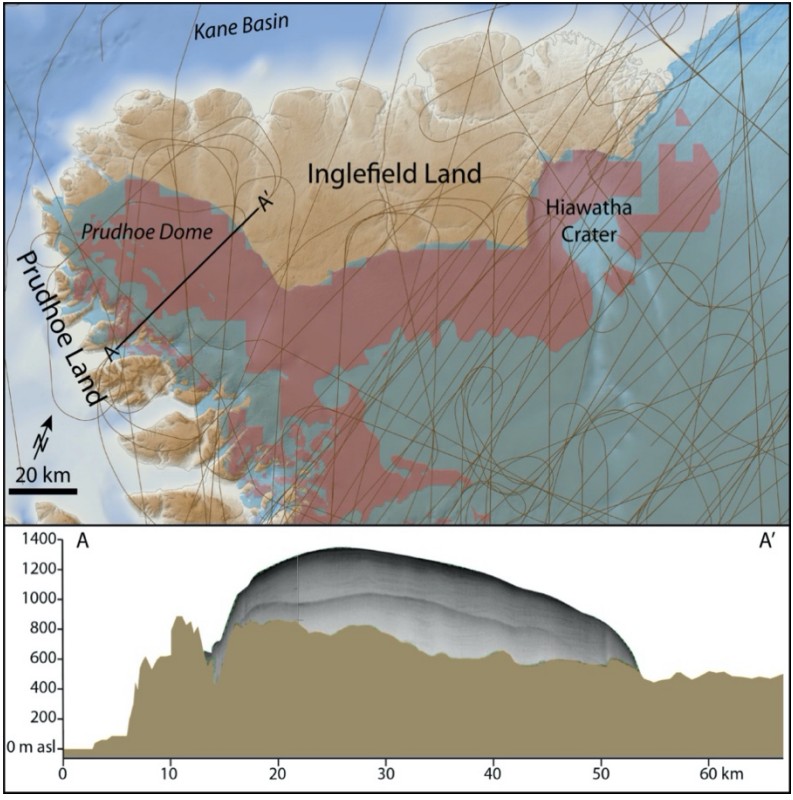

**Figure 5.** Top panel shows areas that meet drilling requirements (shown in red) in NW Greenland. The Greenland Ice Sheet is depicted in light blue, and NASA Operation IceBridge (OIB) flight lines are shown as thin brown lines. Bottom panel shows OIB radar of Prudhoe Dome along A-A', with topography, the ice-sheet bed and the ice-sheet surface from radar collected in 2017; the mid-ice-sheet reflector is the surface multiple. Basemap topography and bathymetry from Morlighem et al. (2017).


### 4.2 North Greenland: Victoria Fjord

Most of North Greenland is dominated by sedimentary rocks of the lower Paleozoic

Franklinian Basin not well suited for providing the hard, quartz-bearing lithologies that work
best for in-situ cosmogenic nuclide analysis (Henriksen et al., 2009). At the head of Victoria
Fjord (Figure 6A), however, Henriksen and Jepsen (1985) describe isolated outcrops of
crystalline basement in otherwise non-quartz-bearing sedimentary-rock-dominated North
Greenland. The crystalline rocks, mostly orthogneiss, comprise several nunataks in Victoria
Fjord, and additionally crop out in the bottom of two valleys between C.H. Ostenfeld and Ryder
glaciers (Figure 6B). The sedimentary formations consist of Neoproterozoic through Silurian
lithologies composed of near-horizontally bedded shale, siltstone and abundant carbonate

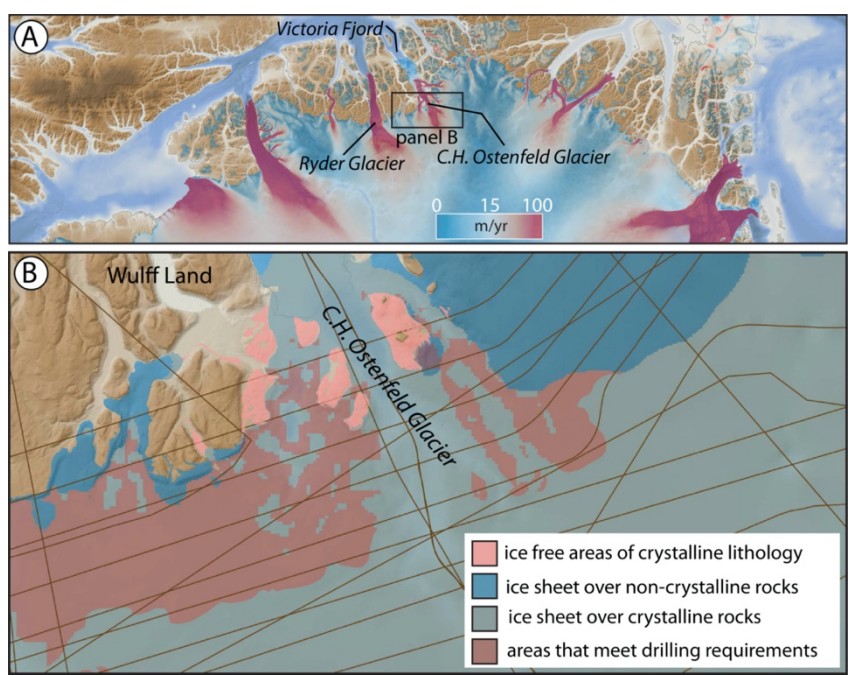

**Figure 6.** A. North Greenland showing ice sheet surface velocity; velocity from Greenland Ice Sheet velocity map from Sentinel-1, winter campaign 2019/2020 [version 1.3]; QGreenland v2.0.  B. Areas that meet drilling requirements with a focus on bedrock lithology: bright blue are areas under the ice sheet with non-quartz bearing lithologies whereas the more muted blue colors depict our estimate of where there are quartz-bearing lithologies at the ice bed. Pink areas are quartz-bearing lithologies beyond the ice margin, and shown in muted red color are the areas that meet the drilling requirements. NASA Operation IceBridge (OIB) flight lines are shown as thin brown lines. Basemap topography and bathymetry from Morlighem et al. (2017).

units. The outcrop pattern is one of crystalline rocks exposed at lower elevations where the
GrIS had eroded away the overlying sub-horizontal sedimentary rocks, or cap rocks. Overall, the
outcrop of these quartz-bearing lithologies is promising for their existence at the ice-sheet bed
south of the ice sheet margin. However, because there are topographic highs along the GrIS
bed south of the margin, blindly drilling into areas that meet the other technical requirements
could lead to encountering cap rocks. For this region, we perform an additional step to estimate
where the bed south of the ice margin may be crystalline vs. sedimentary. To project the
crystalline/cap rock contact southward under the ice sheet, we use the contact between
crystalline rocks and the overlying cap rocks in exposed areas to perform a "3-point problem"
an established method for determining the strike and dip of a plane based on geologic outcrop
patterns. As observed by Henriksen and Jepsen (1985), the contact dips gently to the north, and
the plane that we calculated in ArcGIS confirms this. Our estimation reveals crystalline rocks
outcropping in topographic low areas, and cap rocks outcropping in topographic high areas
(Figure 6B). Our estimated contact is simplistic, as there may be folding and faulting that limit
the accuracy of this extrapolation. However, this solution provides a straightforward estimate
for where crystalline rocks may exist at the ice–bed interface near the head of Victoria Fjord.
Using this information, along with Operation IceBridge flight lines and in combination with
areas that meet the other technical and safety requirements, indicates promising areas to drill
to the southwest of the onset zone of C.H. Ostenfeld Gletscher (Figure 6B).

**4.3 North Greenland: Mylius-Erichsen Land**
In eastern North Greenland there is an ~100-km stretch of ice margin in Mylius-Erichsen
Land (Figure 7) that lies over quartz-bearing sedimentary lithologies of the Proterozoic
Independence Fjord Group (Henriksen et al., 2009). The rocks in this region contain near-
horizontally bedded siltstones, sandstones, and quartzites intruded by Mesoproterozoic

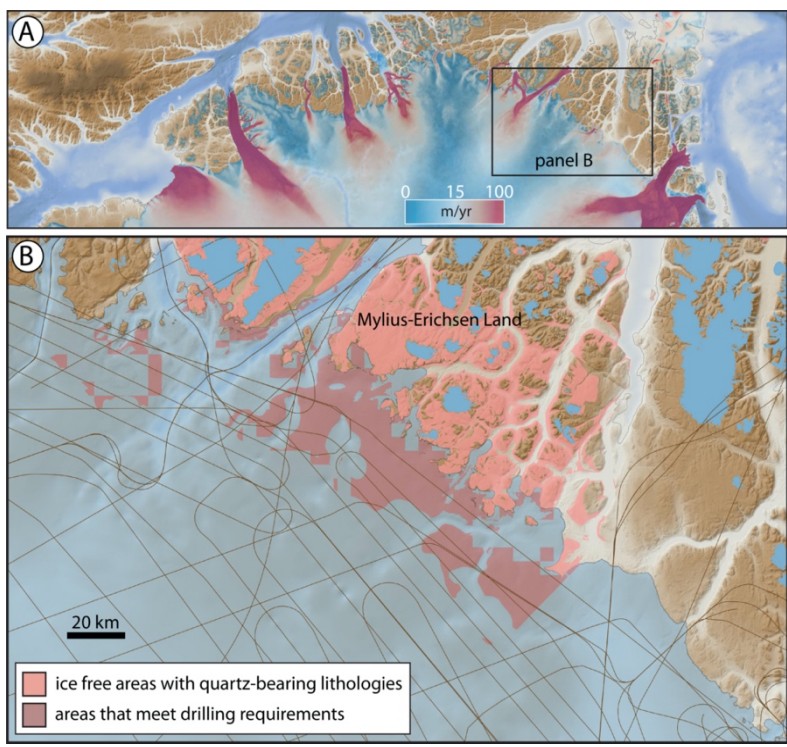

**Figure 7.** A. North Greenland showing ice sheet surface velocity; velocity from Greenland Ice Sheet velocity map from Sentinel-1, winter campaign 2019/2020 [version 1.3]; QGreenland v2.0.  B. Mylius-Erichsen Land showing areas that meet drilling requirements, with pink areas representing quartz-bearing sedimentary formations (with mafic intrusives) beyond the ice margin. NASA Operation IceBridge (OIB) flight lines are shown as thin brown lines. Basemap topography and bathymetry from Morlighem et al. (2017).

dolerite sills, dikes and stocks. Significant portions of the ice sheet in this region meet the
technical and safety requirements for drilling, so the suitability for developing GrIS histories
rests mostly on the likelihood of encountering preferred lithologies. The pattern on the geologic
map and the unit description of the rock formations in the area indicate that the abundance of
mafic intrusions mean that drilling has a reasonable chance of encountering non-quartz-bearing
lithologies. The region has relatively sparse radar lines that cross drilling-suitable areas
compared to other parts of Greenland, narrowing the choices of drill sites that have tight
constraints on ice thickness and bed shape.

**4.4 Northeast Greenland: Dronning Louise Land**

The nunatak region of Dronning Louise Land, Northeast Greenland, contains broad areas

that meet the technical requirements for subglacial drilling (Figure 8). The bedrock geology is
part of the Caledonian fold belt and contains abundant structures that formed during the
Caledonian Orogeny (Ordivician-Devonian) leading to the juxtaposition of crystalline and
younger sedimentary rock formations in a complicated map pattern (Henriksen et al., 2009;
Strachan et al., 2018). The broadest regions that meet the technical criteria and are most
favorable for drilling lie on the western (inland) portion of the coastal mountain ranges. Here,
nunataks exist 25-30 km west of the coastal mountains and provide information on the bedrock
geology most relevant to potential drilling areas. The lithologies are similar to and have been
correlated with the Independence Fjord Group found in Mylius-Erichsen Land. Specifically, the
local unit that comprises inland nunataks (the Trekant Series) consists largely of quartzitic and
feldspathic sandstone and conglomerate with intercalated siltstone and mudstone. Bedrock
mapping in the nearby coastal mountains also reveals Mesoproterozoic dolerite intrusions.
Sparse radar data limits potential drill sites with close constraints on ice thickness and bed
shape. Yet, the region does have potential for tapping into quartz-bearing units and due to its
proximity to the Northeast Greenland Ice Stream, sub-ice cosmogenic nuclide analyses from the
area could yield important constraints on Northeast Greenland Ice Stream history. Finally, the
possibility that high-elevation areas remain glaciated by local ice after inland ice recedes should
not be ignored. Many of the sub-ice drilling targets are >1000 m asl, near twentieth century
snowline elevations. To reduce the chances of drilling a site that is occupied by local ice once
inland ice recedes, one could assess snowline elevation gradients using the presence/absence
of ice caps in peripheral mountains along this coastline. Preliminary analysis shows that
snowline elevations increase inland. Extrapolating these gradients to sub-ice areas suitable for
drilling could help to guide drill site selection by identifying sites with elevations lower than
projected snowline altitudes. In this way, targeting lower-elevation parts of the sub-ice terrain
could be advantageous given the goal of monitoring GrIS history. Additional radar surveys over
key areas would be useful for tightening constraints on ice thickness and bed topography over
the frozen-bedded patches of Dronning Louise Land.

**4.5 East Greenland**
A final place to highlight is central East Greenland, where – similar to Dronning Louise
Land – the GrIS abuts and flows through alpine terrain. The western (inland) flank of these
mountains has dozens of isolated areas that meet the technical requirements of drilling to the
bed (Figure 9). Like the other areas throughout East and Northeast Greenland, the bedrock
geology is highly variable. The headwaters of the Scoresby Sund, Kong Oscar Fjord and Kejser

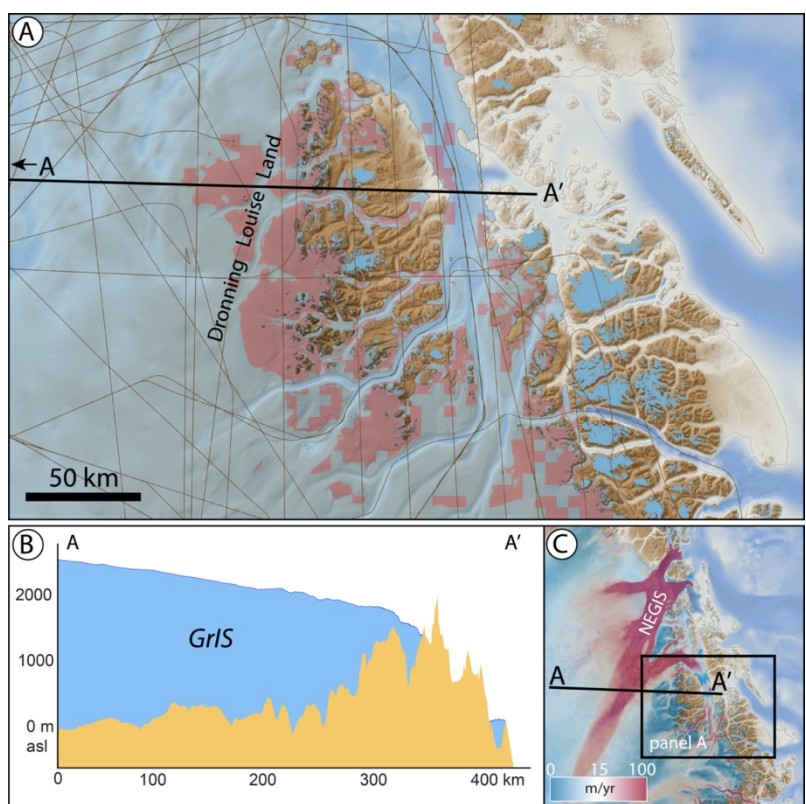

**Figure 8.** A. Dronning Louise Land showing areas that meet drill requirements; NASA Operation IceBridge (OIB) flight lines are shown as thin brown lines. B. Topographic profile of bed and GrIS surface from Bed Machine v3 (Morlighem et al., 2017); cross section line shown in panel C. C. NE Greenland with surface velocity showing NEGIS and location of panel A; velocity from Greenland Ice Sheet velocity map from Sentinel-1, winter campaign 2019/2020 [version 1.3]; QGreenland v2.0. Basemap topography and bathymetry from Morlighem et al. (2017).

Franz Joseph Fjord systems have a complicated geology relating to the Caledonian Orogen,
consisting of Paleoproterozoic crystalline metamorphic and sedimentary formations that are in
turn slightly metamorphosed (Henriksson et al., 2009). In terms of finding quartz-bearing rocks
most suitable for cosmogenic nuclide analysis, the region is heterogeneously made up of
quartz-bearing (e.g., orthogneiss) and non-quartz-bearing formations (various fine-grained
siliciclastic lithologies with occasional mafic intrusions). Inland nunataks provide knowledge of
bedrock geology most proximal to potential drill locations and are largely composed of pelitic
lithologies (e.g., metamorphosed mudstones). Looking closely at nunatak lithologies reveals
some westernmost nunataks of orthogneiss composition, such as inland of J.L. Mowinckel Land
(Figure 9), making the areas in this region that meet the technical requirement promising. A
consideration with the potential drilling locations in central East Greenland, again, is the
likelihood that they are deglaciated with the recession of inland ice, as opposed to retaining
local ice cover. Airborne radar data are also sparse there, so care would be needed to select
sites with the best constraints of ice thickness and bed shape.

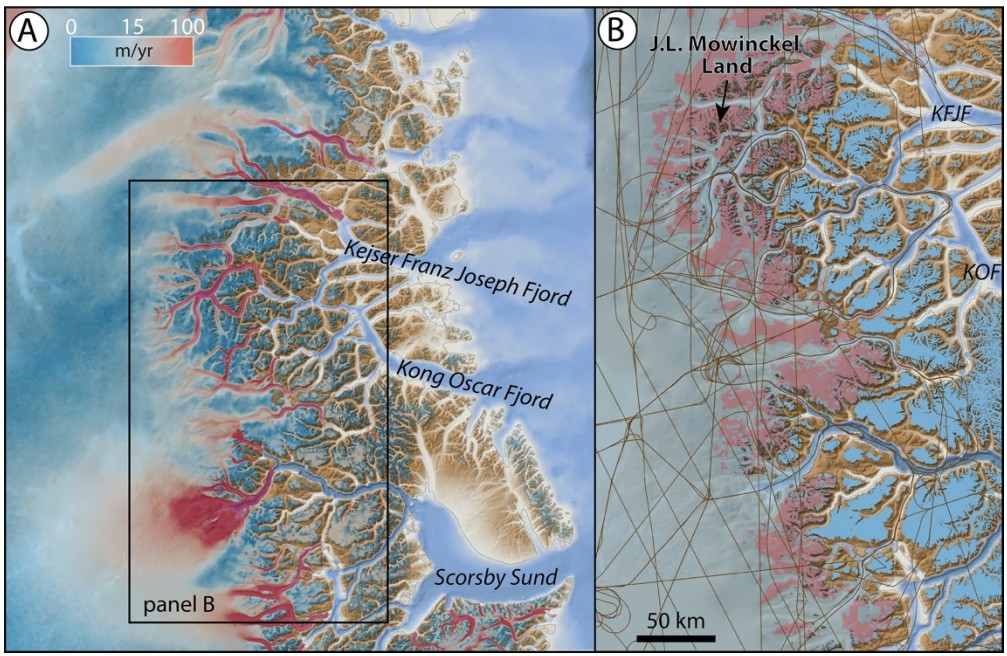

**Figure 9.** A. East Greenland showing GrIS flowing through alpine terrain; surface velocity highlights major outlets; velocity from Greenland Ice Sheet velocity map from Sentinel-1, winter campaign 2019/2020 [version 1.3]; QGreenland v2.0.  B. Portion of East Greenland that includes the most areas that meet the technical requirements of drilling to the bed. Basemap topography and bathymetry from Morlighem et al. (2017).


**5. Conclusions**

The purpose of this study is to identify potential targets for subglacial drilling with

cosmogenic nuclide analysis in mind. We find that only 3.4% of the GrIS is well suited for
cosmogenic-nuclide analysis of bed materials using existing drills available from the US Ice
Drilling Program and highlight five promising locations in northern and eastern Greenland.
Future advances in drill capability, such as the ability to drill through thicker or wet-based ice,
would significantly increase the area available for drilling (Figure 2).
In addition to obtaining drill cores of rock or sediment from the ice sheet bed, samples
of other basal material would also benefit the research community. Basal ice is valuable for (1)
measuring trace gasses to obtain basal ice age (Bender et al., 2010, Yau et al., 2016b), (2)
detrital cosmogenic nuclide analysis of its mineral component (Bierman et al., 2014), and (3)
ancient DNA and biomarkers in organic compounds (Willerslev et al., 2007). Boreholes
themselves that are the product of drilling can be instrumented, resulting in direct
measurements of basal heat flux values that would provide additional constraints on the basal
thermal state of the GrIS (e.g., MacGregor et al., 2022; Colgan et al., 2021) and the history of
the Iceland hotspot (e.g., Rogozhina et al., 2016). Finally, precise sampling at the ice-bed
interface could lead to the discovery of ancient soils that plausibly exist in areas targeted for
drilling that are frozen-bedded for long periods. Such samples may be useful for a variety of
studies including ancient DNA, macrofossil, and biomarker analyses. Additionally, with proper
precautions, the uppermost few millimeters of the bed can be preserved in light-free conditions
and used to measure for luminescence dating, providing an additional chronometer of past ice-
sheet presence/absence (e.g., Christ et al., 2021).
Pairing sub-ice cosmogenic-nuclide analysis with ice-sheet modeling is an important step
(Spector et al., 2018). Ice-sheet model simulations have the ability to scale information from
single drill sites, or transects of sites, to the entire GrIS. Likewise, results from ice-sheet
modeling can help identify which potential drill sites are most sensitive to overall ice sheet
mass balance, thus help to prioritize sites or to assemble a strategically chosen group of sites.
Finally, high resolution ice-sheet models with fine meshes in areas of peripheral mountainous
topography could help with 'local ice survival' issues that could complicate cosmogenic-nuclide
records from areas where alpine topography is smothered by the GrIS.
In our companion paper (Keisling et al., 2022), we use an ensemble of ice-sheet
simulations to illustrate deglaciation styles around the GrIS. The results reveal how much sea
level equivalent the GrIS has lost as each perimeter site becomes ice free, many of which are
reachable by the ASIG drill. The geometry of ice-sheet retreat depends on a number of ice-
sheet model parameters, including climate forcing, lapse rate, model initialization, lithosphere
response, etc. We found that some locations become ice free after a similar amount of ice loss
regardless of the uncertainty in these parameters, whereas other locations experience a range
of ice-cover histories depending on the model parameters. Our results demonstrate how
numerical models can provide another tool to guide site selection by identifying locations
where bedrock-derived evidence for ice-free conditions tells us something concrete about ice-
sheet size and volume. More observational data of past GrIS change, such as cosmogenic
nuclide analyses, will improve the model-based estimates by identifying the deglaciation styles
that are the most realistic, thereby constraining parametric uncertainty. In turn, as models
become more competent, they have the ability to scale single drill-site (or transects of sub-ice
drill sites) information into a broader picture of regional, or whole, GrIS change. As both of
these tools improve, taking an integrated approach offers the greatest potential for leveraging
new breakthroughs into societally relevant information about ice-sheet history and stability.

In summary, we consider this study, and ideally drilling efforts taking place in one or

more of these candidate sites, as only one of several next steps in the exploration of the GrIS
bed and in providing useful data for improving ice-sheet models. We recommend development
of drills that can penetrate thicker ice and potentially ice where the bed is thawed. This could
be done by modifying existing drill technology (e.g., Timoney et al., 2020; Goodge et al., 2021)
or require the development of entirely new drills. Expanding the area of the GrIS available for
subglacial drilling would broaden the range of scientific questions that could be addressed
regarding GrIS history and the range of possible targets. The application of cosmogenic nuclide
analysis of subglacial materials could then move beyond constraining GrIS history during
periods when it is only slightly smaller (~90%) than its present configuration to constraining
times of significant reduction (~<10%). Additionally, there would be more resolving power for a
fuller range of scientific questions, such as what shape the GrIS takes during past interglacials
(Plach et al., 2018; Domingo et al., 2020), where ice dynamics may influence large-scale retreat
(Aschwanden et al., 2019), or where there are packages of subglacial lake sediments (e.g.,
Keisling et al., 2020; Paxman et al., 2021) or unique geologic structures (Kjær et al., 2018;
MacGregor et al., 2019). Evolving drilling techniques and analyses like this pave the way for
targeted exploration of subglacial bed environments, a new frontier in ice sheet and sea level
science.

**Author contribution**

JB led the analysis and writing. CW led geographic-information-system computations. All authors contributed to discussions that resulted in the ideas and analysis presented in this manuscript, and all authors contributed to writing and presentation. We thank Greg Balco and an anonymous referee for important suggestions that helped to improve this paper.

**Acknowledgement**

This work was funded with NSF grant 1933938 (JPB) and 1933927 (NEY, JMS, BAK).

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
