# Peer review of "Drill site selection for cosmogenic nuclide exposure dating of the bed of the"

_EGUsphere, 2022_

## Referee Comment (RC1)

**Where to GreenDrill? Site selection for cosmogenic nuclide exposure dating of the bed of the Greenland Ice Sheet**

This paper focuses on possible locations suitable for drilling through the Greenland Ice Sheet to collect bedrock and/or sub-sediments for cosmogenic nuclide exposure dating. The main purpose of this, is to constrain the glacial history on a longer term scale and especially look into the respond of the Greenland Ice Sheet during past interglacials. Drilling through the Greenland Ice Sheet is a difficult challenge and only few places at the margin of the Greenland Ice Sheet are currently suitable for drilling with the available equipment.

The paper presents the different criteria for drilling which meets the demands of the drilling equipment as well as the geological aspect of valuable site locations. The paper presents the different criteria in a structured and short, clear way and follow up with a discussion on possible places to drill and also present the main selected sites available for drilling and part of this campaign.

The paper was easy to read and with a good presentation that makes the reader clearly understand the idea behind the research proposal and also the chosen localities.

I look very much forward to follow the campaign and the outcome of this project, which I am sure will bring very valuable knowledge to the research community, both regarding the method but also the glacial history of the Greenland Ice Sheet.

Beneath I have listed some areas in which I would like the authors to elaborate on the individual topics together with minor comments on the text and figures.

**General comments**

I would encourage the authors to elaborate on the possible outcome of this project in relation to what they can say about past GrIS history. You drill at the margin so do you only expect to be able to say something about the marginal ice sheet history or would you expect your results, together with previous studies, can open up for a wider GrIS history interpretation, when it comes to the spatial extent? In the abstract and introduction the is mentioning of studies showing an ice sheet wide history, and it would be good if you in a few lines very clearly could state the outcome of this project.

In lines 320-322 you briefly mention the other suitable areas, which meet your criteria, but that where not chosen. Can you elaborate more on why they are not suitable? Maybe give some specific location examples?

Title: This might just be me, but suggesting to change the title to "bed beneath"/"bedrock beneath" instead of "bed of". You also refer to it in this way in lines 33-34; …"cosmogenic nuclides in bedrock from beneath the Greenland Ice Sheet"….

Lines 113-115: Can you elaborate "direct constraints" here? I would consider to delete direct and in general maybe elaborate more. The studies I assume you refer to here place constraints, but as far as I remember do not conclude one unique solution/ice sheet burial/exposure history of measured concentrations in sub-ice material?

Lines 340-343: Can you elaborate on how you will look more into/determine if this area has local ice during past interglacials? How would that affect your modeling and interpretation, would you use a different approach than the other areas etc?

In the conclusion and introduction, you talk about the information retrieved from Camp Century and GISP2 as paradigm shifting and "direct" information, but for me to see they are both "most likely scenario" results, but still with more possible ice sheet histories to fit measured nuclide concentrations? I would consider to make it clearer that there are more than one solutions/result from those studies.

**Minor Comments**

You use both "ice sheet" and "ice-sheet" throughout the text, chose one for consistency

Line 87: Consider to delete "the" before MIS 5e

Line 87: Delete "age"? There is something in the sentence that doesn't make sense

Line 92: "the" sub-ice bedrock exposure age

Line 115: "so far" instead of "thus far"?

Line 122: In the abstract you use "<700 m" and here "~700 m", consider to make consistent

Line 158: This is the second section numbered "3.1"

Line 192: Delete space between "warm-" and "and"? As you have it in line 187

Line 217: This line doesn't read well, do you mean the criterion of being safe, so no air support is needed or the need of air support for transportation during fieldwork?

Line 248: Consider re-phrasing to "in its south, west and central areas" – it feels like something is missing when reading the sentence the way it is now.

Line 262: You change between writing "NASA's Operation IceBridge" and "NASA Operation Ice Bridge" – chose one and make consistent

Line 376: is the "-" after "100" intended?

Lines 282-292: This is up to the authors but it would be great if you could elaborate a bit on why you want the nuclides to be preserved. You want them for the modelling part, but just to elaborate a bit on how you can use "inheritance" and different nuclides, with different half lifes to model past ice sheet extent.

Line 386: Is a "shows" missing after "sparse radar data that"

Line 405: Consider abbreviating Northeast Greenland Ice Stream since you use the abbreviation in the caption for figure 8.

Lines 410-413: Either move to/place instead of lines 403-404 or delete lines 403-404, which also mentions the sparse radar data

Line 464: Consider to delete "to measure"

**Figures**

Figures: "A" or "(A)", chose one and be consistent (same in the rest of the figures)

Figure 1: Suggesting to put the location of NEEM on the map, since it is mentioned several times

Figure 1: Just a suggestion, color either B or C with maybe grey instead of white so they are not both same color

Figure 1: Consider to add a scale

Figure 4: Consider to enlarge the figure or maybe just text in the white box, the text is very small as it is now.

Figure 8: Is its placement wrong? It should be before section 4.5?

Figure 8: This is up to the authors, but I would re-arrange this figure, so (A) and (C) would be in the top panel (with (C) first and then (A)) and (B) would be below in full length.

Figure 9: In (B), do you mean "KFJF" and not "KKJF"?

---

## Author Response (AR1)

GreenDrill paper, response to reviewer comments.

Reviewer 1.

We wish to thank this reviewer for their insights, and we very much appreciate their time. Many comments seem to center around requests to further elaborate in certain areas, below you can find our thoughts for doing so. Also, please note that some of our replies to the other referee's comments apply to these comments, so please check out both of our replies. Thanks!

Review 1 comment: I would encourage the authors to elaborate on the possible outcome of this project in relation to what they can say about past GrIS history. You drill at the margin so do you only expect to be able to say something about the marginal ice sheet history or would you expect your results, together with previous studies, can open up for a wider GrIS history interpretation, when it comes to the spatial extent? In the abstract and introduction the is mentioning of studies showing an ice sheet wide history, and it would be good if you in a few lines very clearly could state the outcome of this project.

Our reply: We like the addition of more specific outcomes. Yes we agree that any drill site using the available drills mentioned are by default in the peripheral areas of today's ice sheet. We feel this is still valuable.

Further, using results from ice-margin exposure dates to constrain the size and shape of the Greenland ice sheet using an ensemble of numerical ice-sheet model simulations is the goal of another study, currently in review. We have added a citation to the pre-print of this study where appropriate in order to emphasize this outcome. However, the focus of this study is on integrating multiple forms of data to find the most suitable locations for subglacial access drilling, so the numerical modeling effort is not included directly here.

The abstract end, we would add: *"Results from cosmogenic nuclide analysis in new sub-ice bedrock cores from these areas would help to constrain dimensions of the Greenland Ice Sheet in the past."*

Review 1 comment: In lines 320-322 you briefly mention the other suitable areas, which meet your criteria, but that where not chosen. Can you elaborate more on why they are not suitable? Maybe give some specific location examples?

Our reply: We explain that we don't emphasize these areas because "most areas are in alpine-style, icefield-type settings, or lie in small areas between outlet glaciers"

The implication is that being smaller target zones to drill into from the ice sheet surface, given existing uncertainties in available datasets (e.g., ice sheet thickness, basal temperature, etc.), they could more easily not pan out than larger patches of the bed that meet drilling requirements. Given the reviewer comment seeking more detail, we would add this statement along those lines: *"In these additional small areas, however, existing uncertainties in available datasets (e.g., ice sheet thickness, basal temperature, etc.) means that drilling there is a bit riskier than in larger patches of the bed that meet drilling requirements."*

Review 1 comment: Title: This might just be me, but suggesting to change the title to "bed beneath"/"bedrock beneath" instead of "bed of". You also refer to it in this way in lines 33-34; ..."cosmogenic nuclides in bedrock from beneath the Greenland Ice Sheet"....

Our reply: Good idea, will change title to replace "of" with "beneath"

Review 1 comment: Lines 113-115: Can you elaborate "direct constraints" here? I would consider to delete direct and in general maybe elaborate more. The studies I assume you refer to here place constraints, but as far as I remember do not conclude one unique solution/ice sheet burial/exposure history of measured concentrations in sub-ice material?

Our reply: We see a difference between direct/indirect, versus, say, how closely a dataset might constrain ice sheet change. We take indirect data as those such as ice-rafted debris, sediment flux or sediment geochemical records as being indirect on ice sheet size since there are other factors in addition to ice margin position that control their fluctuations. See referee 2's comments on this; they clearly agree. Direct, on the other hand, we take here as cosmogenic nuclide inventory in bedrock, which directly relates to past ice sheet size (presence v absence at a site of data analysis). Cosmogenic nuclide measurements in sediments, which can be transported to a site where today they are measured, may lie somewhere in between (further discussed elsewhere in our reply). All this said, we also agree of course that there are multiple interpretations of sub-ice cosmogenic nuclide inventories (could be one or many periods of ice-free exposure that result in a particular concentration). There are, however, some conclusions less open to interpretation, such as overall cumulative periods of time that a site has been ice free, or cumulative time that a site has been buried by overlying ice.

We already write on line 99: *"Although alternative histories are possible, the results point to significant ice loss in Greenland within the Quaternary, and likely within the last 1.1 Myr." {note Referee 2 advises us to remove "likely."}*

We would like to clarify that there may not be a single solution, yet there still is an important and direct constraint on cumulative ice sheet presence/absence at a site (compared to most proxies for minimal ice sheet configurations). Based on both referee comments (collectively), we would now write: *"Although alternative histories are possible, the results lead to an important conclusion: an almost entirely absent ice sheet in Greenland within the last 1.1 Myr. Furthermore, these types of data directly constrain past ice-sheet configurations, unlike marine sediment records from adjacent seas that provide indirect evidence."*

Review 1 comment: Lines 340-343: Can you elaborate on how you will look more into/determine if this area has local ice during past interglacials? How would that affect your modeling and interpretation, would you use a different approach than the other areas etc?

Our reply: While we cannot distinguish whether ice covering the Prudhoe Dome site is binned as "local" or "ice sheet" ice, how that ice is categorized is less important than determining if any ice was in these peripheral settings at all during the entire Holocene, or Last Interglacial, and so on. Via use of multiple isotopes, one should be able to simply determine the Holocene history, and likely could explore MIS 5e history versus earlier exposure. Add-on analyses like luminescence methods could help elucidate pre-Holocene exposure timing more finely. Ice sheet models could be paired with cosmogenic nuclide information to help visualize whether or not peripheral high elevation areas persist for long durations after inland ice has receded or not. This is addressed further in our reply to Referee 2.

Review 1 comment: In the conclusion and introduction, you talk about the information retrieved from Camp Century and GISP2 as paradigm shifting and "direct" information, but for me to see they are both

"most likely scenario" results, but still with more possible ice sheet histories to fit measured nuclide concentrations? I would consider to make it clearer that there are more than one solutions/result from those studies.

Our reply: This is an important comment, and we are thankful that it has been repeated in comments made by this referee. By now we have addressed this comment since it echoes one made above about "direct constraints." Note that we also discuss added interpretations of cosmogenic nuclide data in sediments in section 3.5.

Review 1 comment: Minor Comments
You use both "ice sheet" and "ice-sheet" throughout the text, chose one for consistency

Our reply: In English, a compound modifier should be hyphenated if it comes before the word it modifies. When ice sheet is used as an adjective, it is intentionally hyphenated (eg "…ice-sheet model."); and when it is used as a noun ("…under the ice sheet."), it is not hyphenated.

Line 87: Consider to delete "the" before MIS 5e
Our reply: Done.

Line 87: Delete "age"? There is something in the sentence that doesn't make sense
Our reply: Done.

Line 92: "the" sub-ice bedrock exposure age
Our reply: We think it makes more sense to leave "the" out as it refers to not one example site but the bed age more generally.

Line 115: "so far" instead of "thus far"?
Our reply: Subjective.

Line 122: In the abstract you use "<700 m" and here "~700 m", consider to make consistent
Our reply: Done.

Line 158: This is the second section numbered "3.1"
Our reply: Oops. Thank you.

Line 192: Delete space between "warm-" and "and"? As you have it in line 187
Our reply: Actually, it is more correct to add a space on line 187, haha. So now both lines are consistent with each other.

Line 217: This line doesn't read well, do you mean the criterion of being safe, so no air support is needed or the need of air support for transportation during fieldwork?
Our reply: Done.

Line 248: Consider re-phrasing to "in its south, west and central areas" – it feels like something is missing when reading the sentence the way it is now.
Our reply: Done.

Line 262: You change between writing "NASA's Operation IceBridge" and "NASA Operation

Ice Bridge" – chose one and make consistent
Our reply: Done.

Line 376: is the "-" after "100" intended?
Our reply: Yes intended.

Lines 282-292: This is up to the authors but it would be great if you could elaborate a bit on why you want the nuclides to be preserved. You want them for the modelling part, but just to elaborate a bit on how you can use "inheritance" and different nuclides, with different half lifes to model past ice sheet extent.
Our reply: We agree that we can clarify that without cosmogenic nuclides preserved in the first place, then one of the main reasons to obtain sub-ice bedrock is already out the window. So, we add to the middle of the paragraph (line 289) the following sentence: "At sites of erosive sub-glacial conditions, cosmogenic nuclides would be largely absent, hence removing one of the main reasons for obtaining sub-ice bedrock samples."

Line 386: Is a "shows" missing after "sparse radar data that"
Our reply: We agree this is a little unclear, would change "data" to "lines": "The region has relatively sparse radar lines that cross drilling-suitable areas."

Line 405: Consider abbreviating Northeast Greenland Ice Stream since you use the abbreviation in the caption for figure 8.
Our reply: We favor using acronyms as little as possible. Makes more sense in a figure caption which is a condensed piece of information.

Lines 410-413: Either move to/place instead of lines 403-404 or delete lines 403-404, which also mentions the sparse radar data
Our reply: Thanks for pointing out the repetition. We would remove "The regional has relatively sparse radar data coverage, suggesting that" from Line 410/4011.

Line 464: Consider to delete "to measure"
Our reply: Agree.

Review 1 comment: Figures
Figures: "A" or "(A)", chose one and be consistent (same in the rest of the figures)
Our reply: Agree.

Figure 1: Suggesting to put the location of NEEM on the map, since it is mentioned several Times
Our reply: Agree.

Figure 1: Just a suggestion, color either B or C with maybe grey instead of white so they are not both same color
Figure 1: Consider to add a scale
Our reply: We'd opt to leave scale bars off, and will leave both Fig 1B and Fig 1C white.

Figure 4: Consider to enlarge the figure or maybe just text in the white box, the text is very small as it is now.

Our reply: Agree.

Figure 8: Is its placement wrong? It should be before section 4.5?
Our reply: Will fix, this would change during type setting anyway.

Figure 8: This is up to the authors, but I would re-arrange this figure, so (A) and (C) would be in the top panel (with (C) first and then (A)) and (B) would be below in full length.
Our reply: Agree, this makes good sense.

Figure 9: In (B), do you mean "KFJF" and not "KKJF"?
Our reply: Agree, great catch.

Reviewer 2, Greg Balco.

Hi Greg, thanks for the detailed and insightful review, we really appreciate the time you took. Please note that some of our replies to the other referee's comments relate to some of your comments, so please check out both of our replies. Thanks again!

Review 2 comment: This paper is an extremely clear explanation of the reasoning behind how one would locate drill sites aimed at applying subglacial bedrock exposure dating to learn about Greenland Ice Sheet response to past climate warming. As it is the paper is quite clear and can be published in approximately its present form. It could be improved by adding some more sophisticated discussion in some sections, as described below.

One initial thing is that I would leave 'Greendrill' out of the title. It's a US-specific project name and it doesn't really make any sense to anyone who doesn't know about the project. Of course the project is described within the paper, but it makes the title unnecessarily incomprehensible before reading. Also, the reasoning described in the paper would apply to any subglacial drilling project in Greenland, not just this one. So I'd make the title more general to just indicate that the paper is about site selection criterial for subglacial bedrock recovery drilling.

Our reply: We agree and suggest instead *"Drill site selection for cosmogenic nuclide exposure dating of the bed beneath the Greenland Ice Sheet"*

Item 1. The one thing that is really missing in this paper that could be improved is a better discussion of assessing the sensitivity of potential observables at a drill site to ice sheet size and dynamics. The authors allude to this near line 277 with the statement that drill sites "should be robust monitors of past ice sheet margin change." However, this statement doesn't really mean anything by itself. From the rest of the paragraph I take this to mean that there should be a strong correlation between the ice thickness or ice margin position at the drill site and some parameter of broader interest, for example total ice sheet volume or the position of the ice margin at a location of interest like a major outflow glacier. Drill site selection for this purpose in Antarctica has focused on this idea by using ice sheet models to develop theoretical transfer functions between overall ice sheet volume and ice thickness at a candidate drill site -- a good example of this is in Spector, 2018, Figure 2 (https://tc.copernicus.org/articles/12/2741/2018/tc-12-2741-2018.pdf), and this approach has been used in several other proposals and planned projects. This is valuable because it can both identify sites where the ice thickness at the site is likely to be related to ice volume, and also because it can identify

sites where the response to expected variability or specific collapse scenarios is accessible within the design depth range of a particular drill system. This approach is also discussed in a simple way in the Schaefer et al. paper about the GISP2 bedrock, which uses model results to show that the majority of the ice sheet must be gone for the site to be exposed.

Our reply: We appreciate very much the expertise of this referee, thanks much for sharing your experience and years of thinking about these problems. First, we think that in Greenland, ice presence/absence at most drill site location scales with ice volume - and ice margin position. (Exceptions are perhaps in some peripheral mountains where increased precipitation during interglacials may lead to local ice survival during times when the main ice sheet experiences more negative mass balance.) Of course, however, the relationship between ice presence/absence at a particular site and ice volume may have different slopes (sensitivities) at different potential drill sites. As the referee points out, this relationship can be characterized with ice sheet models. We feel this is a bit beyond the scope of this "suitable sites" paper, but additionally, our group is exploring this and is presented in a companion paper that is currently in review elsewhere led by co-author Keisling. In that submitted paper, we explore what referee Balco is getting at, and we generate pixel-by-pixel map of the sensitivity that each potential drill site location is to the volume loss of the ice sheet as a whole. We have added references to the publicly available pre-print of this companion paper where appropriate in the text.

Unfortunately, the exact approach to transfer functions between ice thickness at a site and total ice volume that is in the Spector paper is probably not as useful in marginal areas of Greenland because of the ice margin geometry: in Antarctica, ice thickness variations are manifested as changes in the exposed height of steep-sided interior nunataks, but ice-free areas in Greenland are for the most part marginal horizontal plateau surfaces that are progressively exposed or covered by ice margin migration across a flat surface. Of course even in this geometry the extent of exposed land surface has to be related to the inland ice thickness in a general way, but it's much more complicated. It's also possible that the ice margin position on plateaus in certain colder areas might be related to local mass balance, so if mass balance is less positive during colder times, ice margin position at some places might even behave inversely to total ice volume. Figuring out the appropriate transfer function could be a hard problem for most likely Greenland drill sites, and it's really going to be the central challenge of interpreting whatever results are eventually revealed by the drilling project. I don't think developing this in detail is a necessity for publication of this paper, but I think it needs to be discussed in a more quantitative way.

Our reply: We agree that "developing this in detail is a necessity for publication of this paper." Discussing this in a more quantitative way is a next step, and is started in the Keisling et al companion paper that focuses on the ice-sheet modeling side of this.

Specifically, I suggest expanding this part of the paper into a separate section entitled 'Relating drilling results to overall ice sheet geometry,' or something of that nature, that contains the points that (i) one wants a drill site where some kind of transfer function can be established between the observable results and the overall condition of the ice sheet, and (ii) some ideas and geometric considerations for how to establish that transfer function.

Our reply: Great idea. Given our pending companion paper and its focus on this topic, we would rather pass on this particular suggestion, as great as it is, and instead leave it to our other paper.

The issue of whether or not ice domes could or could not occur on highlands separately from the ice sheet is also important in this context, and could also be discussed in a more quantitative way. For example, there are many highland plateaus in North Greenland that have ice domes on them, and many that do not. What is the difference? This is clearly important in understanding what happens at a drill site when the main ice sheet retreats. Is there a minimum size or elevation necessary to sustain an ice dome? Do they only occur in areas with particular temperature/precip conditions? Of course, this paper isn't going to completely do this analysis, but it could give some pointers for how to approach the problem.

Our reply: Another excellent point. When choosing sites that are inherently around the ice sheet periphery areas for our funded work, we performed some rudimentary analysis of equilibrium-line altitude patterns/gradients.  Referee Balco is keen in observing that some peripheral mountains are glaciated, others not. One can use this pattern, along with topographic data, to show that ELAs rise inland. In selecting sites for our funded work, we projected these gradients inland using bedmachine elevation data and targeted ice-sheet bed sites below the local projected equilibrium line altitudes. There are caveats of course, firstly it is today's ELA, not one of the past, and it ignores changes in precipitation patterns in a scenario with a reduced ice sheet. But, nevertheless, it points us in the right direction.

Toward the end of the section 4.4, we already include this text: *"Finally, the possibility that high-elevation areas remain glaciated by local ice after inland ice recedes should not be ignored. Many of the sub-ice drilling targets are >1000 m asl, near twentieth century snowline elevations."*

To expand a bit in light of the good points made by the referee, we propose tack on the following: *"To reduce the chances of drilling a site that is occupied by local ice once inland ice recedes, one could assess snowline elevation gradients using the presence/absence of ice caps in peripheral mountains along this coastline. Preliminary analysis shows that snowline elevations increase inland. Extrapolating these gradients to sub-ice areas suitable for drilling could help to guide drill site selection by identifying sites with elevations lower than projected snowline altitudes."*

A final point in this area is that the paper could do a better job of articulating that an array of boreholes from one area is a lot better than one borehole. If you just have one borehole, the site was either exposed, or not, in the past. If you have an array of boreholes at different distances from the ice margin, you may be able to say where the ice margin was at a certain time, or at least establish the cumulative frequency distribution of ice margin extent.

Our reply: As the reviewer knows, this is the strategy in our funded work.  However, while yes, the concept is absent from this manuscript, we point out that this paper is not a proposal to do the work, nor is it meant to outline the work that our co-author team is funded to do - it simply lays out the considerations for where one can drill using ASIG and Winke drills. We could fold in a recommendation for this strategy – but again that would just be our opinion for how to use our "suitable areas for drilling" findings in one's future scientific mission.  Really those details should be left to groups who wish to drill and use the analysis in this paper to guide their work.  In any case, we think there is a way to mention the power of a transect without it being necessarily "our recommendation"… We propose combining this in a paragraph where we also address the referee's other comment about the time trade-off with drilling few 700 m holes vs. many 100 m holes. So see our proposed paragraph farther down in this reply.

Item 2. Another aspect of the discussion that could be improved has to do with the importance of subsurface core vs. surface samples. This is briefly alluded to near line 232 and near line 260, but this doesn't clearly make the important point, which is that if you only have a surface sample you can't tell the difference between a short period of exposure at the surface and a long period of exposure beneath some layer of snow, ice, or rock cover. With a depth profile extending more than about a meter below the surface, you can tell the difference between these two things. Again, this is already described at length in lots of places (e.g., Schaefer 2016 again), so many folks should know it already, but this paper should make this point clearly.

Related to this point is that this paper needs to clarify the difference between drilling into bedrock and sediment. Again, this is alluded to in the line-260 region, but the text doesn't clearly state the important point, which is that if bedrock shows evidence of exposure, the exposure must have been at exactly that location. If you have sediment that shows evidence of exposure, the exposure could have taken place at a different location and the sediment transported (as is somewhere between possible and likely for the Camp Century samples). The paper should make this point clearly. Another related point, of course, is that if the sediment hasn't moved since the last exposure, it's equivalent to bedrock. This is likely relevant for some of the potential drill sites on plateaus that are probably always under ice divides with low flow velocity, so loose surface sediment or saprolite at these locations is possibly equivalent to bedrock for exposure-dating purposes.

Our reply: This is an excellent comment. We agree that it is worthwhile to expand the bedrock vs. sediment discussion, and to clarify how bedrock cores are more advantageous than surface samples. We thank the referee for adding fresh eyes on this - and opening our eyes to this omission.  In fact, when looking at our section "3.4 (now 3.5) Cosmogenic nuclides and subglacial geology", we found that it needs to be overhauled a bit to be better streamlined and reduce some redundancy. We propose adding new text to prior text to clarify bedrock vs. sediment:

*"Sampling from a bedrock substrate has advantages over samples from sediment deposits, although cosmogenic nuclide measurements from both are informative. Sediments beneath ice sheets are more easily eroded, deformed, entrained, transported and re-deposited than bedrock. Thus, cosmogenic nuclide concentrations from the sediment grains themselves, which have a transport and deposition history, are more complicated to interpret than those in bedrock. Furthermore, cold-based ice that flows atop sediment sections can more easily erode a sediment surface (via entrainment processes) than in bedrock substrates. Thus, not only is a cosmogenic signal in sediments derived from each individual grain's exposure and burial elsewhere (that are later amalgamated into a single deposit), but the ice-bed itself may not represent a prior land "surface." Thus, sediment samples could be from an arbitrary depth below a paleo-surface. The depositional environment of sediment is also important. If ice overlies a fluvial sediment sequence, then the cosmogenic nuclide inventory is highly likely to have a complicated genesis, and thus a more complicated interpretation. On the other hand, if the sediment is saprolite or regolith, and largely formed in-situ, then its cosmogenic nuclide inventory likely would be more straight forward to interpret. In any case, for targeted sub-GrIS cosmogenic nuclide campaigns, the highest priority sites are those where non-erosive ice rests directly on quartz-bearing bedrock."*

…and adding new text clarifying the advantage of bedrock depth profiles vs. surface samples only:

*"Cosmogenic nuclide analyses made in a depth profile below the ice-bed interface yield important information. Measurements in a rock core spanning a meter or more, for example, can allow one to easily identify whether or not the current ice-bed interface has been eroded and/or covered by snow, ice*

*or sediment for long durations (e.g. Schaefer et al., 2016). On the other hand, one cannot determine with surface-only samples whether a surface has been impacted by minor erosion and/or burial by snow, ice or sediment. Thus, analysis of bedrock cores is most important for elucidating ice sheet histories from cosmogenic-nuclide inventories. Furthermore, cores spanning several meters and including depths dominated by muon production have the added advantage of constraining orbital-scale term exhumation histories (e.g., Balter-Kennedy et al., 2021)."*

Item 3. The section on 'Considerations for drilling' could benefit from some discussion of the time-depth tradeoff in drilling operations. In a field season you can (maybe) drill a couple of 700-meter holes, or a significantly greater number of 100-m boreholes. So restricting sites to < 700 m ice thickness establishes feasibility, but not necessarily optimality.

Our reply: This is another good point. Is it the job of this author team and in this paper to advise how one choses to spend their field season? Maybe it is… We have added a new paragraph to the end of section 3.6 (the last section in the "considerations for drilling" part of the manuscript.

*"Finally, additional considerations relating to field season planning could lead to meeting scientific goals most efficiently. Multiple drill cores along transects (even including sites beyond the present – ephemeral – position of the ice-sheet margin) could boost confidence in constraining past ice-sheet dimensions through time. For example, a site that is presently covered by 100 m of ice may have been ice-free during the Holocene, whereas a 400-m-thick site farther inland was not; thus, one could better constrain the position of the ice margin during the middle Holocene. Additionally, using the ASIG Drill, there may be enough time in a field season to acquire one or two drill cores from thicker ice sites (e.g., 500-700 m), versus obtaining many drill cores in a single season from ~100m-thick sites using the Winke Drill. The optimal sampling strategy depends on several factors that relate to a particular scientific objective."*

Item 4. Not related to the scientific merit of the paper, but I found the conclusions disappointing because they essentially restate motivational material that should have been in the introduction (e.g., all of 466-482). The discovery that subglacial exposure dating is an important thing to be doing is not an outcome of this work -- we knew it already, as evidenced by the fact that the GreenDrill proposal was successful. What the reader is hoping for here is more a statement of what was learned from the work done in the paper: e.g., the screening process described in the paper yielded a large number of candidate drill sites, but they are in fairly restricted areas of the ice sheet, and they present some geometric challenges to understanding the relationship between ice sheet size and drill site exposure. Following discussion above, I would probably highlight that the next real challenge is to establish how the exposure, or lack thereof, of the plateaus that are the most likely candidate drill sites is related to the broader condition of the ice sheet.

Our reply: Fair point. We removed the conclusion pre-amble that, as referee Balco points out, is a re-hash of the paper introduction. We propose leaving in, however, some of the other text. What might seem obvious (and maybe boring) to the reviewer (and maybe to our co-author team), it is actually a small subset of scientists who think about these things - we feel that the conclusion like that which currently exists would benefit students and an array of cryosphere scientists who haven't thought as much about this type of work (e.g., paleo, cosmogenic nuclide methods).

On the other hand, we appreciate the good ideas here about folding in the need for future work, largely ice-sheet modeling (some of which is ongoing), that when married to the CRN analysis from sub-glacial

drilling, provide a much more complete picture of whole-ice sheet history and can likewise address "local ice problems." We would add the following to the conclusion as new paragraphs:

*"Pairing sub-ice cosmogenic-nuclide analysis with ice-sheet modeling is an important step (Spector et al., 2018). Ice-sheet model simulations have the ability to scale information from single drill sites, or transects of sites, to the entire GrIS. Likewise, results from ice-sheet modeling can help identify which potential drill sites are most sensitive to overall ice sheet mass balance, thus help to prioritize sites or to assemble a strategically chosen group of sites. Finally, high resolution ice-sheet models with fine meshes in areas of peripheral mountainous topography could help with 'local ice survival' issues that could complicate cosmogenic-nuclide records from areas where alpine topography is smothered by the GrIS."*

*"In our companion paper (Keisling et al., 2022), we use an ensemble of ice-sheet simulations to illustrate deglaciation styles around the GrIS. The results reveal how much sea level equivalent the GrIS has lost as each perimeter site becomes ice free, many of which are reachable by the ASIG drill. The geometry of ice-sheet retreat depends on a number of ice-sheet model parameters, including climate forcing, lapse rate, model initialization, lithosphere response, etc. We found that some locations become ice free after a similar amount of ice loss regardless of the uncertainty in these parameters, whereas other locations experience a range of ice-cover histories depending on the model parameters. Our results demonstrate how numerical models can provide another tool to guide site selection by identifying locations where bedrock-derived evidence for ice-free conditions tells us something concrete about ice-sheet size and volume. More observational data of past GrIS change, such as cosmogenic nuclide analyses, will improve the model-based estimates by identifying the deglaciation styles that are the most realistic, thereby constraining parametric uncertainty. In turn, as models become more competent, they have the ability to scale single drill-site (or transects of sub-ice drill sites) information into a broader picture of regional, or whole, GrIS change. As both of these tools improve, taking an integrated approach offers the greatest potential for leveraging new breakthroughs into societally relevant information about ice-sheet history and stability."*

Other than that, it's great. I enjoyed reading it and the figures are excellent, except that Figure 3 should be bigger. Just because of the nature of the result the reader is trying to look at small areas in a big ice sheet. Use a whole page for this figure. Otherwise, great job with the figures.

Our reply: Agree, bigger figure 3.

Additional minor items:

Lines 66-79. This section could be greatly simplified simply by noting that marine records are indirect evidence no matter what. The only direct evidence that an ice sheet was not in a certain place is evidence from under the ice sheet that something else was there.

Our reply: Referee 1 asked for clarity on direct v. indirect evidence of past ice sheet change. We opt not to remove text from this paragraph, but do clarify some language, as stated in response to Ref 1 comments, we would now write: *"Although alternative histories are possible, the results lead to an important conclusion: an almost entirely absent ice sheet in Greenland within the last 1.1 Myr. Furthermore, these types of data directly constrain past ice-sheet configurations, unlike marine sediment records from adjacent seas that provide indirect evidence."*

Line 101. 'Likely' is incorrect. The Schaefer et al. data require surface exposure no earlier

than 1.1 Ma.

Our reply: Thank you, will make change.